# Post-Exercise Shifts in the Hemato–Biochemical Profile of Unacclimatized Camels (*Camelus dromedarius*)

**DOI:** 10.3390/ani15213061

**Published:** 2025-10-22

**Authors:** Mohammed A. Al-Badwi, Emad M. Samara, Khalid A. Abdoun, Ahmed A. Al-Haidary

**Affiliations:** Department of Animal Production, College of Food and Agriculture Sciences, King Saud University, P.O. Box 2460, Riyadh 11451, Saudi Arabia

**Keywords:** animal physiology, bleeding time, electrolyte balance, osmolality, recovery kinetics, serum enzymes

## Abstract

**Simple Summary:**

Camels are well known for their ability to live in deserts, but little is understood about how their blood chemistry changes after hard work in hot conditions, especially if the animals are not trained. In this study, healthy young camels were asked to perform a 90 min outdoor exercise session at midday heat, and their blood was tested before and at several times after exercise. The results showed that red blood cells, which carry oxygen, remained largely unchanged. In contrast, the ability of blood to clot became stronger right after exercise but returned to normal within a few hours. Levels of salts in the blood, such as sodium, potassium, and phosphate, shifted in clear patterns: sodium rose and fell within the first six hours, while potassium and phosphate stayed low for up to two days. Proteins and sugars in the blood also changed temporarily, and muscle-related enzymes increased, especially lactate dehydrogenase, showing signs of muscle stress. Overall, the camels needed at least 48 h to return to balance. These findings are important because they provide practical guidance for veterinarians and owners on how long camels need to rest after strenuous work in the heat to protect their health and welfare.

**Abstract:**

Exercise-unacclimatized dromedary camels regularly perform strenuous work in desert heat; however, their short-term hematologic and biochemical recovery is not well defined. In this prospective repeated-measures experiment, seven healthy bulls underwent a standardized 90 min outdoor exercise bout, with blood sampled before exercise and at 0, 3, 6, 24, and 48 h of recovery. The analytical panel included hematology, primary hemostasis, electrolytes, osmolality, protein fractions, metabolites, and serum enzymes. Red-cell indices remained stable, indicating minimal erythrocyte mobilization, while bleeding time shortened sharply at 0 h and normalized by 3 h. Sodium and osmolality followed a biphasic pattern with an early rise at 3 h, a nadir at 6 h, and partial rebound by 24 h, whereas potassium and phosphate stayed depressed from 6 to 48 h. Proteins and glucose showed transient changes, and muscle-associated enzymes, especially lactate dehydrogenase, peaked early before declining. These findings demonstrate that camels tolerate combined exercise and heat stress but require up to 48 h to re-establish biochemical balance. The recovery timeline provides a clinically relevant framework for sampling, welfare monitoring, and management of work–rest cycles in desert environments.

## 1. Introduction

Dromedary camels routinely operate under thermal loads that would overwhelm most domestic species. Their survival in arid environments is supported by unique adaptations that stabilize circulating volume, preserve effective perfusion, and sustain homeostasis under extremes of heat and dehydration [1,2]. Even so, when high ambient temperatures coincide with exercise, the resulting combined stress imposes rapid demands on fluid–electrolyte balance, cardiovascular distribution, and cellular integrity. In exercise-unacclimatized animals, such challenges can rapidly exceed their natural compensatory capacity.

Prior work in camels has documented that exercise and heat exposure provoke measurable shifts in blood composition, particularly in electrolytes, glucose, and muscle-associated enzymes, while red-cell indices typically remain near baseline [3,4,5]. Race-like exertion has been linked to post-exercise increases in osmolality, transient stress hyperglycemia, and elevations in muscle-associated enzymes. Unlike horses, where splenic autotransfusion produces marked hematologic surges, camels generally demonstrate modest changes in RBC, Hb, and Hct, consistent with minimal mobilization of erythrocytes during work [3,4]. Hematologic interpretation is further complicated by biological covariates such as age and sex, which shape reference ranges and confound comparisons at single time points [6]. Primary hemostasis is also labile, where bleeding time shortens immediately after exertion and typically normalizes within hours, implying a transient activation and shear-related priming of platelets, rather than sequestration or marginating behavior within vessels, consistent with sympathetic catecholamine effects [7].

Despite this foundation, the scientific literature remains fragmented in both temporal scope and analytic breadth. Actually, several studies capture only brief windows of ≤3–24 h or restrict analysis to narrow marker sets. For instance, early osmolality and hormone changes after maximal exercise, or transient cardiac troponin-I (cTnI) spikes that return to baseline by 24 h, without mapping multi-variable recovery across the first days post-exercise in unacclimatized animals [8,9]. Such constraints leave clinicians and physiologists without an integrated reference for when hematologic and biochemical signals should peak, invert, or normalize.

The present experiment was therefore designed to address this gap. Using a prospective repeated-measures design, we combined conventional laboratory methods with time-course visualization of deviations from baseline over the first 48 h following outdoor field exercise under desert heat. By mapping explicitly the trajectories of hematology, electrolytes, protein fractions, metabolites, and enzymes in unacclimatized camels, this experiment provides a clinically relevant, time-stamped framework for interpreting recovery, planning training or workload, and guiding welfare-oriented interventions in hot climates.

## 2. Materials and Methods

All procedures involving animals were reviewed and approved by the Institutional Research Committee of King Saud University, Riyadh, Saudi Arabia (process number: KSU-SE-24-38), ensuring adherence to ethical standards for the welfare and humane treatment of research animals.

The experiment employed a prospective repeated-measures design with seven clinically healthy bull dromedary camels, aged 4–6 years and weighing between 350 and 450 kg. Each animal served as its own control. Camels were individually housed in shaded open pens (4 m × 5 m) and provided ad libitum access to fresh water and feed. The diet consisted of roughage in the form of alfalfa hay and Rhodes grass (*Chloris gayana* spp.), offered twice daily at 06:00 and 17:00 h, supplemented with a commercial pelleted ration (Al-Wafi, Arabian Agricultural Services Co., Riyadh, Saudi Arabia). All animals were confirmed clinically healthy upon veterinary examination, with no history of recent illness, or prior training. No sedation was administered during the experimental procedures.

Animals underwent a standardized 90 min outdoor exercise bout conducted at mid-day (12:00–14:00) under direct solar exposure. Exercise was performed on a circular sandy track at an average speed of 14 km·h^−1^ (3.91 m·s^−1^), corresponding to an intermediate gait between pacing and galloping, as described previously [10,11]. The camels were unmounted and guided by experienced handlers who maintained a steady pace through timed laps and consistent verbal cues, ensuring comparable effort across all animals. The workload was sufficient to elicit measurable physiological stress while ensuring animal safety. Effort, in terms of both pace and duration, was standardized across animals to the extent permitted by field logistics, with the same handlers supervising the entire exercise. In addition, water was withheld from the animals during the experimental session, beginning 2 h prior to exercise and remained unavailable until the completion of the exercise bout. Immediately upon completion, the 0 h blood sample was collected within 10–15 min, after which camels were returned to their shaded pens for recovery monitoring. It is worthwhile to mention as well that all seven animals successfully completed the exercise protocol and all sampling sessions, with no exclusions occurred. Selection was limited to adult males to minimize hormonal variation associated with the breeding cycle and to ensure behavioral consistency during handling and treadmill work. Age range (4–6 years) was selected to represent physiologically mature but non-geriatric camels, where cardiorespiratory and metabolic responses are considered stable. A CONSORT-style flow diagram (see Appendix A) was included to summarize animal enrollment, inclusion criteria (n = 7; clinically healthy adult males aged 4–6 years; no exclusions), and complete sampling participation.

To characterize both antecedent and recovery profiles, jugular venous blood samples were obtained at six time points: 2 h pre-exercise (−2 h; designated PRE), immediately after exercise (0 h), and during recovery at 3, 6, 24, and 48 h. Ambient dry-bulb temperature (Ta, °C) and relative humidity (RH, %) were recorded at each sampling using two HOBO H-08 Pro Series data loggers (Onset, Bourne, MA, USA) mounted adjacent to the pens at ~2 m above ground. Before the experiment, both loggers were cross-checked in parallel under identical conditions to confirm agreement within the manufacturer’s tolerance range (±0.3 °C for temperature and ±2% for humidity), and the mean of their concurrent readings was used for subsequent analysis. Measurements at the start and end of the exercise bout, as well as at each recovery sampling, documented the prevailing biometeorological load. A schematic of the sampling timeline and venipuncture points is shown in Figure 1.

Jugular venipuncture was performed using a consistent side within each animal whenever feasible to minimize procedural variability. Specifically, blood was drawn from the left jugular vein in all animals across all time points to ensure procedural consistency and reduce side-related variability. Approximately 8 mL of whole blood was collected into potassium-EDTA tubes for hematology and plain tubes for serum biochemistry. Both tube types were commercial vacuum-based collection systems, ensuring controlled draw pressure and minimizing hemolysis risk. Samples were transported in a chilled insulated carrier and delivered immediately to the laboratory. The field site was located approximately 2 km from the analytical laboratory, and transport plus initial processing were completed within one hour of collection to ensure sample stability. In fact, red blood cell count (RBC), hemoglobin concentration (Hb), and hematocrit (Hct) were analyzed within 1 h of collection using an automated hematology analyzer (Beckman Coulter, Miami, FL, USA).

In addition, primary hemostasis was assessed as bleeding time (BT), which was measured following the standardized procedure described by Samara et al. [7] in camels. Briefly, two minor uniform incisions (5 mm long × 2 mm deep) were made using a disposable bleeding device (Organon Teknika Corp., Durham, NC, USA) within a pre-shaved 5 cm × 5 cm area on the lateral aspect of the shoulder and hip regions. Blood was collected at 20 s intervals onto filter paper without touching the wound, and the mean time to cessation of bleeding was recorded using a digital stopwatch. All measurements were performed by the same trained operator to avoid inter-observer variability. To assess procedural consistency, three animals were retested one week later under identical conditions, showing <6% variation in bleeding time. Ambient temperature (36–38 °C) and handling procedures were standardized across all tests.

For serum samples, they were allowed to clot at ambient temperature for no longer than 30 min, centrifuged at 3000 rpm for 10 min at 5 °C, and the supernatant serum was aliquoted into sterile cryotubes to prevent repeated freeze–thaw cycles. All sera were stored at −20 °C and analyzed in a single batch within 7 days of collection. The interval between venipuncture and final analysis did not exceed 10 days. Serum biochemical constituents were determined using standard procedures. Serum osmolality was measured by freezing-point depression osmometry (Advanced Instruments, Model 3320, Norwood, MA, USA). Although the osmometer inherently reports results in milliosmoles per kilogram of water (mOsm/kg H_2_O), equivalent values in milliosmoles per liter (mOsm/L) are presented to maintain consistency with previous camel physiology and clinical studies that have traditionally used volumetric units. Electrolytes (Na^+^, K^+^, Cl^−^, Ca^2+^, PO_4_^3−^) were quantified using validated colorimetric methods on a semi-automated chemistry analyzer (RX Monza, Randox Laboratories, Crumlin, UK). Serum phosphate (PO_4_^3−^) was expressed exclusively in milligrams per deciliter (mg/dL), in accordance with the manufacturer’s calibration and prevailing veterinary reporting standards. Protein fractions were determined by the biuret method for total protein, bromocresol green for albumin, and by difference for globulin. Metabolites (glucose, blood urea nitrogen, and creatinine) were assayed enzymatically, whereas enzyme activities [aspartate aminotransferase (AST), alanine aminotransferase (ALT), lactate dehydrogenase (LDH), and alkaline phosphatase (ALP)] were measured by kinetic methods on the same analyzer platform at 37 °C. Serum enzyme stability during frozen storage was verified in a pilot test using duplicate aliquots (n = 3 camels) stored for 1 week at −20 °C and reanalyzed post-thaw, showing less than 4% deviation in AST and LDH activities. Calibrators and internal quality controls were run before each assay series. The intra- and inter-assay coefficients of variation were <3.5% and <5.0%, respectively. All biochemical values were standardized, and reference physiological ranges for adult dromedary camels were incorporated in Table 1 for contextual interpretation.

Data were analyzed using linear mixed-effects models to account for repeated measurements across time within the same animal. Time (six levels: PRE, 0, 3, 6, 24, and 48 h) was treated as a fixed effect, and animal was included as a random effect to account for individual variability. The general statistical model was defined as: (*Y_ij_* = *μ* + *T_i_* + *A_j_* + *ε_ij_*), where *Y_ij_* represents the observed value for the *j*th animal at the *i*th time point, *μ* is the overall mean, *T*_*i*_ denotes the fixed effect of time, *A*_*j*_ the random effect of animal, and *ε_ij_* the residual error. Analyses were performed in SAS (version 9.4, PROC MIXED) using a first-order autoregressive covariance structure to model correlations among repeated measures within animals. The full model structure, including fixed and random effects and covariance specification, was verified for each variable. Global time effects were evaluated using the Type III tests of fixed effects, and significance was declared at *p* < 0.05. When a significant main effect of time was detected, post hoc pairwise comparisons were performed using Tukey’s adjustment for multiple testing, and results were considered significant when *Pₐdj* < 0.05. For each variable, η^2^ and Cohen’s *f* were computed from one-way repeated-measures ANOVA, and post hoc observed power values were calculated using F-test estimations (n = 7 animals × 6 time points). The resulting power values ranged from 0.17 to 1.00 across the twenty variables, confirming high sensitivity (≥0.80) for twelve variables and moderate for four additional ones (see Appendix A). These quantitative indices, together with adjusted *p*-values (*Pₐdj*) and standardized effect sizes (Cohen’s *d* with 95% confidence intervals), are reported in Table 1 to facilitate transparent interpretation of both statistical and biological relevance. Model adequacy was confirmed by examining residual plots for normality and variance homogeneity and by assessing overall model fit across time points. Mean ± standard error (SE) values are reported in Table 1. Additionally, individual-level spaghetti plots were generated for each of the twenty variables, overlaid with mean trajectories, to illustrate within-animal recovery profiles. To enhance interpretability, deviations from baseline (Δ = value at time–PRE) were computed and displayed as multi-panel plots showing the direction and magnitude of change across all variables.

## 3. Results

Biometeorological measurements (Ta and RH) fluctuated across the sampling schedule. At PRE, 0, 24, and 48 h, Ta and RH values were statistically indistinguishable (*p* ≥ 0.05), with mean Ta of 39.16 ± 0.32 °C and RH of 15.24 ± 0.12%. At 3 h, Ta decreased (36.02 ± 0.39 °C, *p* < 0.05) while RH increased (21.74 ± 0.12%, *p* < 0.05). The most pronounced divergence was observed at 6 h, where Ta reached its lowest value (29.41 ± 0.34 °C, *p* < 0.05) and RH its highest (31.40 ± 0.13%, *p* < 0.05). These patterns confirm that animals were consistently exposed to hot conditions, with peak environmental load evident at PRE/0 h and again at 24–48 h.

Turning to hematology, variables remained largely stable throughout the protocol (Table 1). Red-cell indices (RBC, Hb, Hct) fluctuated narrowly around PRE values, with modest short-lived decrements immediately after exercise (0 h) and partial rebound by 3–6 h. None of these fluctuations yield an effect (*p* ≥ 0.05). In contrast, primary hemostasis demonstrated a marked but transient response, where BT shortened at 0 h (*p* < 0.05), returned to baseline by 3 h (*p* ≥ 0.05), and remained near PRE values for the remainder of the recovery period.

Electrolyte–osmotic variables showed clear temporal structuring (Table 1). Sodium increased above baseline at 3 h (*p* < 0.05), decreased below baseline at 6 h (*p* < 0.05), and then increased again by 24 h (*p* < 0.05). Osmolality mirrored this biphasic trajectory, with a transient elevation immediately after exercise followed by gradual normalization during recovery. Although this pattern did not reach statistical significance after Tukey adjustment (*Pₐdj* = 0.078), the moderate effect size (*d* = 0.62) suggests a biologically meaningful osmotic response to exertion. Potassium values remained consistently below baseline from 6 through 48 h (*p* < 0.05), and phosphate concentrations were likewise depressed compared to PRE at all post-exercise time points (*p* < 0.05). Chloride and calcium showed modest fluctuations without consistent directionality (*p* ≥ 0.05).

Additionally, protein fractions displayed staggered temporal kinetics (Table 1). Total protein was highest at 0 h (*p* < 0.05) and remained mildly elevated at 24 h. Albumin reached peak levels between 6 and 24 h (*p* < 0.05), whereas globulin rose immediately at 0 h (*p* < 0.05) before declining toward baseline.

Likewise, metabolites followed distinct trajectories however within baseline variability (*p* ≥ 0.05), where glucose was higher at 0–3 h with attenuation thereafter, BUN trended downward across recovery, and creatinine showed small late negative deviations. On the other hand, enzymes were the most responsive variables (Table 1). In fact, AST (*p* < 0.05) increased with a peak at 3 h, ALT rose modestly (*p* ≥ 0.05), and ALP was lower (*p* ≥ 0.05) than baseline (most notably at 24 h) before partial recovery by 48 h. For LDH, its activity tended to rise at 3 h post-exercise but the change was not statistically significant (*Pₐdj* = 0.383). Nevertheless, the moderate effect size (*d* = 0.55) may reflect a transient increase in muscular enzyme release related to short-term exertional stress.

The accompanying post hoc power analysis (Appendix A) confirmed that variables showing statistically significant time effects, particularly Na^+^, K^+^, PO_4_^3−^, Ca^2+^, AST, LDH, and ALP, were supported by high observed power (≥0.80), ensuring that detected differences reflected genuine physiological shifts rather than sampling variability. Conversely, parameters with lower observed power (e.g., RBC, Hb, Cl^−^, ALT) displayed stable trajectories, indicating biological constancy rather than underpowered inference.

The individual-level trajectories illustrated in Figure 2 reveal distinct temporal dynamics among hematological and biochemical variables following exercise. Red-cell indices (RBC, Hb, Hct) showed narrow oscillations around baseline values, with minor early declines and full recovery by 6 h, confirming erythrocytic stability. Bleeding time displayed a steep fall immediately post-exercise, followed by rapid normalization at 3 h and sustained stability thereafter. Sodium and osmolality followed synchronized biphasic trends, peaking at 3 h, reaching their lowest point at 6 h, and partially recovering by 24 h, whereas potassium declined steadily from 6 h onward, remaining subnormal up to 48 h. Phosphate mirrored this pattern, showing persistent depression during the recovery window. Calcium exhibited mild late increases without clear directionality, while chloride remained largely unchanged. Total protein and albumin rose gradually to maxima between 6 h and 24 h, whereas globulin spiked briefly at 0 h before returning to baseline. Glucose increased transiently at 0–3 h, followed by normalization, while BUN and creatinine decreased modestly after 6 h. Enzymatic profiles were more responsive, where AST and LDH peaked sharply at 3 h, ALT showed small non-significant fluctuations, and ALP exhibited a transient decline at 24 h.

Figure 3 illustrates these trajectories as deviations from baseline (Δ from PRE) across all variables. Stability of RBC, Hb, and Hct is reflected in bars clustered near zero at all time points, while the bleeding-time panel displays a pronounced negative bar at 0 h, followed by a return to the zero line at 3 h and thereafter. The biphasic behavior of sodium and osmolality is rendered as early positive bars at 3 h, negative bars at 6 h, and renewed positive bars at 24 h. The sustained depression of potassium and phosphate appears as consecutive negative bars from 6 through 48 h. A staggered temporal pattern in protein fractions is evident in the immediate positive bar for globulin at 0 h versus later positive bars for albumin (6–24 h) and the early elevation in total protein (0 h, with a smaller rise at 24 h). For metabolites, glucose panels show early positive bars (0–3 h) with smaller bars thereafter, BUN shows a sequence of small negative bars, and creatinine shows minor late negative bars. Enzyme panels highlight amplitude, where LDH exhibits the largest positive bars (maximal at 3 h), AST shows a distinct positive bar at 3 h, ALT shows smaller positive bars, and ALP displays a negative bar at 24 h with re-approach toward zero by 48 h.

## 4. Discussion

The performed time-resolved analysis herein was designed to address the limitations noted earlier in the Introduction; namely, that most previous camel studies often restricted themselves to short observation windows or narrow sets of biochemical markers, leaving a fragmented picture of recovery. Actually, our approach hews to that plan and allows both the magnitude and the direction of temporal change to be read at a glance.

The obtained findings showed that red-cell indices were quiescent. The near-isometric behavior of RBC, Hb, and Hct, even in the presence of transient hyperproteinemia, points to a short-lived plasma volume shift rather than the splenic autotransfusion seen in horses [14]. Camels therefore appear to rely less on acute erythrocyte mobilization and more on metabolic and ventilatory strategies to secure oxygen delivery during exertion. This interpretation is consistent with studies demonstrating that in dromedaries, circulatory stability under stress is maintained primarily by fluid balance and osmotic adjustments, not by large fluctuations in red-cell mass [15,16,17,18,19,20]. Shifting from oxygen-carrying capacity, the early response of platelet-dependent hemostasis changed promptly. BT shortened immediately after exercise and normalized within three hours, which accords with earlier reports showing that exertion and dehydration briefly enhance platelet function in camels [7]. The transient shortening most likely reflects reversible platelet activation and vascular shear-related priming rather than sequestration or marginating behavior, aligning with sympathetic catecholamine-driven responses documented across species [21,22]. While the effect was temporary, it highlights the importance of periprocedural timing when evaluating hemostasis around strenuous work. Comparatively, protein fractions separated over time, with mild hyperproteinemia at 0 h followed by mild hyperalbuminemia at 6–24 h and an earlier mild globulin crest. Taken alongside largely unchanged red-cell indices, this profile favors brief hemoconcentration with later oncotic re-equilibration rather than a cell-mass phenomenon [3].

Notably, electrolyte–osmotic variables showed the clearest temporal structure. Sodium and osmolality followed a biphasic course, with elevations (hypernatremia/hyperosmolality) at three hours, a nadir (hyponatremia/hypo-osmolality) at six hours, and partial recovery by 24 h. These shifts are consistent with rapid water–salt redistribution regulated by vasopressin and aldosterone [8,23]. The biphasic pattern likely reflects transient hemoconcentration during exertion followed by osmotic equilibration during recovery, independent of immediate drinking behavior, as water access was controlled until the completion of sampling. Because individual drinking volumes were not quantified, this remains a limitation when extrapolating to field conditions where camels have unrestricted access to water. We also cannot exclude minor contributions from the timing of water intake or diurnal variation in sampling. Prior studies in camels have documented an early post-exercise rise in osmolality followed by a decline toward baseline [4,8], which also indicates that acclimation status, training discipline, and sampling horizon can modulate these trajectories. Overall, our findings represent time-dependent physiological adjustments to exercise rather than statistically confirmed differences, as the adjusted *p*-values exceeded 0.05. Nonetheless, the moderate effect sizes and consistent directional changes indicate a plausible biological trend involving transient hemoconcentration and mild enzyme leakage during early recovery.

In contrast to these biphasic dynamics, other electrolyte trends were persistent. First, hypokalemia from 6 to 48 h was not mirrored in all performance settings, plausibly reflecting β2-adrenergic drives and aldosterone-linked renal handling, compounded by cellular uptake during ATP resynthesis [24,25]. Despite post-race hyperkalemia having been reported elsewhere [4], this may suggest context-dependence (e.g., workload, acclimation, ration) that cautions against universal claims. In that cited investigation, camels were exercised at racing velocities (~20–25 km·h^−1^) and under trained conditions, whereas the present animals were untrained and exercised at a moderate field pace of 14 km·h^−1^, as specified in the Methods. The lower muscular workload and shorter depolarization duration likely reduced potassium efflux from active muscle fibers, accounting for the absence of post-exercise hyperkalemia in this cohort. The second trend is the sustained hypophosphatemia across recovery, which may index high-flux phosphate trapping in muscle with remobilization lag [18,19], although mild dilution remains a competing explanation. These persistent deficits carry practical implications, since they may restrict energy metabolism and cellular recovery if workloads are repeated before equilibrium is restored.

Turning to metabolic functions, the observed mild stress hyperglycemia during the first three hours was in line with counter-regulatory hormone effects and hepatic glucose output described previously in racing camels [4,26]. Additionally, nitrogenous solutes did not show the rises one might anticipate; the observed small downward trajectories in BUN and creatinine are actually compatible with early rehydration and improved renal perfusion, though inter-individual variance can dilute mean change [20,23]. Comparably to these metabolic stabilities, enzyme kinetics told a different story and most strongly implicated skeletal muscle. In fact, serum AST peaked earlier and waned, ALT rose modestly, ALP dipped transiently, and LDH exhibited the largest and most durable hyperenzymemia. This constellation of enzyme changes is consistent with exercise-induced sarcolemmal permeability rather than hepatocellular injury, in line with reports across athletic species [3,14]. In camels engaged in competitive racing, myocardial stress is best assessed through cTnI, which rises immediately after exertion and declines toward baseline within twenty-four hours [9]. Enzyme monitoring using creatine kinase myocardial band (CK-MB), when combined with cTnI assays, therefore provides a more complete picture of tissue-specific load during recovery.

Methodological considerations are important, and we acknowledge three constraints. First, group averages may conceal individual differences, particularly because this study involved only adult male camels aged 4–6 years. The restriction to males was intentional to avoid sex-related hormonal fluctuations that could influence hematological and enzymatic parameters. Nevertheless, this design limits direct generalization to females, juveniles, or aged animals, whose responses to exercise may differ in magnitude or kinetics [6]. Future work incorporating sex and age as model factors would help refine predictive accuracy and ecological applicability. Post hoc precision analysis confirmed that, despite the modest sample size (n = 7), the experiment retained adequate statistical sensitivity to detect medium-to-large effects in most variables (power ≥ 0.80). This supports the validity of the observed temporal trajectories, although mixed-effects extensions could further capture inter-individual heterogeneity in larger cohorts. Second, field conditions such as hydration status, feeding schedules, solar exposure, and diurnal timing introduced additional variability that could not be fully controlled under desert field conditions. Third, although water access was carefully managed during the experiment to eliminate immediate post-exercise drinking effects, individual intake beyond the 48 h observation period was not recorded. This constraint should be considered when extrapolating osmotic and sodium responses to free-ranging or farm-managed camels with unrestricted water availability. Nevertheless, the consistency of direction and sequence across hemostatic, osmotic, and enzymatic domains supports the interpretation of these patterns as physiological adjustments rather than pathological responses. With these caveats in mind, a useful cross-check comes from our own thermophysiology experiment in exercise-unacclimatized dromedaries, where body temperature and cardiorespiratory indices peaked immediately after exercise, thermal gradients intensified between three and six hours, and recovery was largely achieved within twenty-four to forty-eight hours (Samara et al., unpublished observations). The mapping of biochemical fluctuations onto thermal recovery strengthens our view that recovery in camels is asynchronous across subsystems but ultimately convergent toward stability.

Finally, these findings can be embedded into applied welfare and management frameworks. Time-stamped biochemical profiles provide a reference for sampling strategy, welfare assessment, and return-to-work decisions [27]. They also intersect with molecular insights into camel heat tolerance [28] and correspond with veterinary sports medicine strategies for exercise management in hot environments [23]. Therefore, a pragmatic implication follows directly from our data. Clinically, the first six to twenty-four hours after exercise should be interpreted with awareness that hemostasis resolves quickly, sodium and osmolality fluctuate dynamically, potassium and phosphate remain depressed, and enzymes peak early before tapering. Future studies should extend monitoring to 72–96 h, integrate acid–base and endocrine indices, and contrast acclimatized with unacclimatized cohorts. Such work will advance our mechanistic understanding to guide targeted interventions that accelerate recovery while safeguarding adaptive balance, with emphasis on time-structured recovery and welfare-oriented deployment in camels.

## 5. Conclusions

Exercise-unacclimatized dromedary camels tolerate combined thermal and exertional loads, but recovery of hemato–biochemical balance requires up to forty-eight hours. Hematologic indices remain essentially stable, and BT shortens immediately post-exercise, indicating a transient increase in primary hemostasis, before returning to normal by about 3 h. Sodium and osmolality follow a biphasic pattern, showing an early overshoot with a nadir around 6 h, whereas potassium and phosphate stay depressed from 6 to 48 h. Muscle-associated enzymes rise promptly (most notably LDH, with a smaller AST increase) and then decline, with early peaks that may persist through the first day. The temporal sequence observed reflects physiological adjustments rather than pathology, differing from the faster recovery profiles of other species, and underlining the camel’s adaptive strategy of conserving water while redistributing thermal and metabolic loads instead of pursuing rapid normalization. Accordingly, for programs targeting performance and welfare, this timeline offers a benchmark for planning and assessing acclimation status and workload. Clinical sampling and management should be scheduled with reference to key windows; bleeding-time changes in the first hours, sodium and osmolality fluctuations at 3–6 h, hypokalemia and hypophosphatemia from 6 to 48 h, and transient enzyme elevations within the first 24 h. These time-stamped profiles, statistically supported by post hoc power and effect-size analyses, provide a validated framework for clinical evaluation, welfare monitoring, and return-to-work decisions under desert conditions.

## Figures and Tables

**Figure 1 animals-15-03061-f001:**
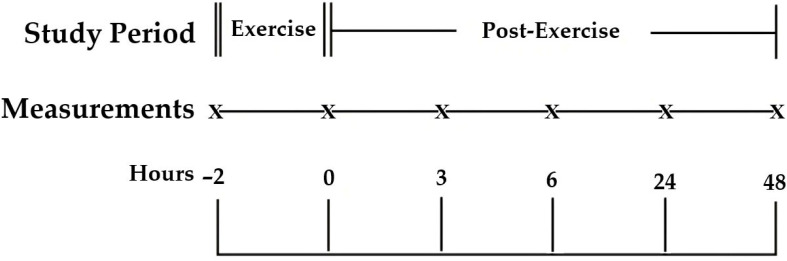
Study design and sampling schedule. Schematic of the repeated-measures protocol. Venipuncture time points are marked by “X”: −2 h (pre-exercise phase), 0 h (immediately post-exercise), and 3, 6, 24, and 48 h (post-exercise recovery). The exercise bout occurred between the −2 h (pre-exercise sample, used as PRE baseline in analyses) and the 0 h. Environmental conditions were recorded during every sampling time.

**Figure 2 animals-15-03061-f002:**
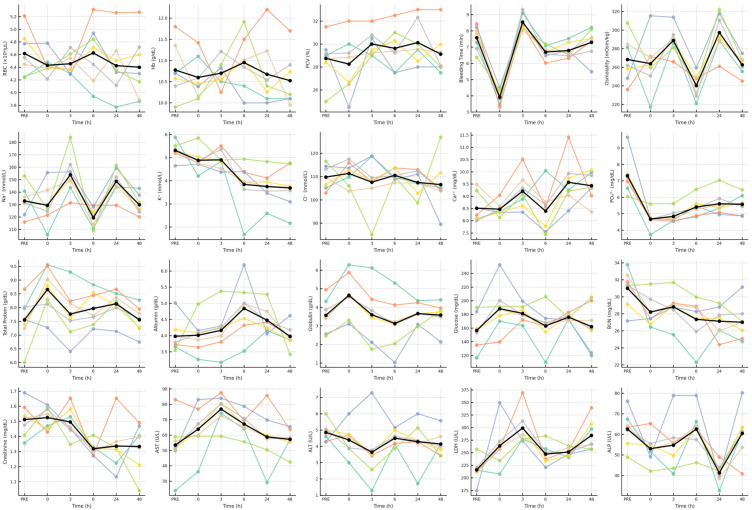
Time-series spaghetti plots of hematological and biochemical variables in heat-stressed unacclimatized camels following exercise. Each subplot shows individual animal trajectories (colored lines) and the group mean (black line with circular markers) for twenty variables measured at six sampling times (PRE, 0, 3, 6, 24, and 48 h). The plots highlight inter-individual variability and coordinated recovery across hematological, osmotic, renal, hepatic, and metabolic systems, illustrating the progressive re-establishment of biochemical equilibrium and the sequential physiological adjustments during post-exercise recovery under desert heat.

**Figure 3 animals-15-03061-f003:**
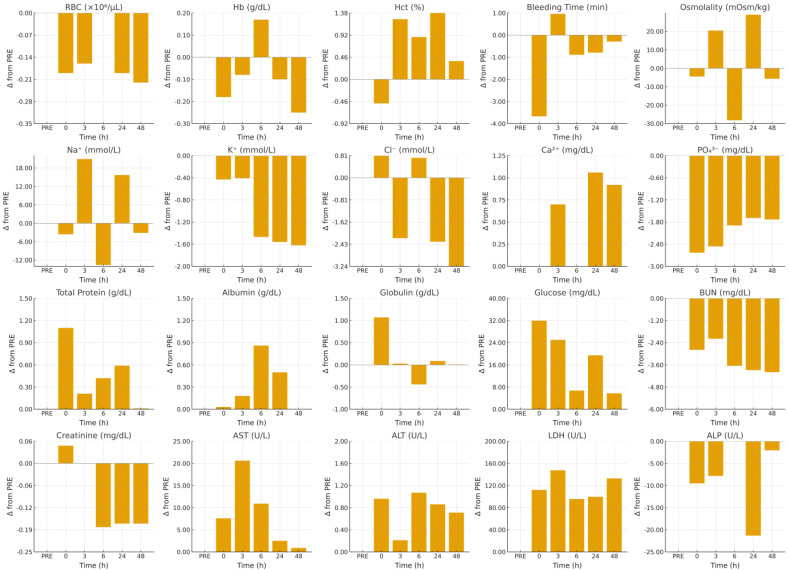
Multi-panel vertical bar charts show, for each measured hematological and biochemical variables, the time-course deviations from baseline (Δ from PRE) over 48 h of recovery after exercise in unacclimatized heat-stressed dromedary camels. Positive bars indicate values above baseline; negative bars indicate decrements.

**Table 1 animals-15-03061-t001:** Hematological and biochemical variables at baseline and during 48 h recovery after a single bout of exercise in unacclimatized heat-stressed dromedary camels.

Variables ^1,2^	Time (h)						SEM	*Pₐdj* (Tukey)	Cohen’s d (95% CI)
	PRE	0 h	3 h	6 h	24 h	48 h			
**Hematology**									
** *RBC* (×10^6^ µL^−1^)**	4.62	4.43	4.46	4.63	4.43	4.40	0.12	0.941	0.11 (–0.42 to 0.65)
** *Hb* (g dL^−1^)**	10.78	10.60	10.70	10.95	10.68	10.53	0.28	0.860	0.09 (–0.48 to 0.63)
** *Hct* (%)**	28.75	28.25	30.00	29.63	30.13	29.13	0.96	0.670	0.18 (–0.39 to 0.72)
** *Bleeding Time* (min)**	7.58 ^ab^	3.91 ^c^	8.54 ^a^	6.69 ^b^	6.79 ^b^	7.29 ^ab^	0.44	0.003	1.26 (0.41 to 2.00)
**Biochemistry**									
** *Osmo* (mOsm L^−1^)**	268.30	263.90	288.80	240.20	297.40	262.70	14.20	0.410	0.33 (–0.27 to 0.91)
** *Na^+^* (mmol L^−1^)**	132.90 ^bc^	129.40 ^b^	153.90 ^a^	119.40 ^c^	148.70 ^ab^	129.80 ^bc^	10.00	0.047	0.92 (0.03 to 1.71)
** *K^+^* (mmol L^−1^)**	5.31 ^a^	4.88 ^ab^	4.90 ^ab^	3.84 ^b^	3.75 ^bc^	3.69 ^bc^	0.37	0.028	1.03 (0.19 to 1.82)
** *Cl^−^* (mmol L^−1^)**	109.80	111.30	107.60	110.50	107.40	106.60	5.40	0.980	0.04 (–0.52 to 0.61)
** *Ca^2+^* (mmol L^−1^)**	8.51	8.47	9.21	8.39	9.57	9.43	0.41	0.210	0.45 (–0.15 to 1.04)
** *PO_4_^3−^* (mg dL^−1^)**	7.30 ^a^	4.67 ^b^	4.84 ^b^	5.41 ^b^	5.61 ^b^	5.57 ^b^	0.46	0.012	1.12 (0.32 to 1.88)
** *Total Protein* (g dL^−1^)**	7.55	8.65	7.76	7.97	8.14	7.56	0.26	0.320	0.41 (–0.18 to 0.97)
** *Albumin* (g dL^−1^)**	3.98	4.01	4.16	4.84	4.48	3.98	0.32	0.392	0.56 (–0.09 to 1.21)
** *Globulin* (g dL^−1^)**	3.57	4.64	3.60	3.13	3.66	3.58	0.36	0.270	0.49 (–0.17 to 1.14)
** *Glucose* (mg dL^−1^)**	156.40	188.40	181.40	163.20	175.90	162.20	15.20	0.641	0.27 (–0.34 to 0.89)
** *BUN* (mg dL^−1^)**	31.00	28.20	28.80	27.40	27.10	27.00	1.20	0.250	0.36 (–0.25 to 0.95)
** *Creatinine* (mg dL^−1^)**	1.50	1.55	1.50	1.32	1.33	1.33	0.08	0.260	0.38 (–0.23 to 0.95)
** *AST* (IU L^−1^)**	56.30 ^b^	63.90 ^b^	76.90 ^a^	67.20 ^b^	58.80 ^b^	57.20 ^b^	3.70	0.014	1.05 (0.21 to 1.83)
** *ALT* (IU L^−1^)**	3.43	4.39	3.64	4.50	4.29	4.14	0.80	0.774	0.15 (–0.48 to 0.75)
** *LDH* (IU L^−1^)**	151.60	263.70	299.10	247.20	251.30	284.60	34.30	0.392	0.43 (–0.18 to 1.04)
** *ALP* (IU L^−1^)**	62.60	53.00	54.70	62.60	41.30	60.50	4.80	0.310	0.51 (–0.11 to 1.15)

^1^ Values are expressed as mean ± SEM (n = 7 animals per time point). *Pₐdj* = Tukey-adjusted probability controlling for multiple testing across time points. Within each row, means bearing different superscript letters (a–c) differ significantly at *p* < 0.05 according to the Tukey HSD test. Cohen’s *d* (95% CI) denotes standardized effect size for the primary contrast (PRE vs. post-dehydration or rehydration times). ^2^ Reference physiological ranges for adult dromedary camels were derived from studies of healthy animals [6,7,12,13]. These correspond to: red blood cells (RBC, ×10^6^ µL^−1^) 7.5–12.6; hemoglobin (Hb, g dL^−1^) 11.1–16.5; haematocrit (Hct, %) 24.1–35.5; bleeding time (min) 3–7; osmolality (Osmo, mOsm L^−1^) 285–310; sodium (Na^+^, mmol L^−1^) 132–152; potassium (K^+^, mmol L^−1^) 3.5–5.2; chloride (Cl^−^, mmol L^−1^) 95–115; calcium (Ca^2+^, mmol L^−1^) 2.1–3.1; phosphate (PO_4_^3−^, mg dL^−1^) 4.8–8.4; total protein (TP, g dL^−1^) 5.0–8.0; albumin (g dL^−1^) 2.5–5.0; globulin (g dL^−1^) 2.0–3.5; glucose (mg dL^−1^) 56–158; blood urea nitrogen (BUN, mg dL^−1^) 20–50; creatinine (mg dL^−1^) 0.8–2.0; aspartate aminotransferase (AST, IU L^−1^) 24–374; alanine aminotransferase (ALT, IU L^−1^) 8–37; lactate dehydrogenase (LDH, IU L^−1^) 300–900; and alkaline phosphatase (ALP, IU L^−1^) 50–187, which align with the expected physiological limits of adult dromedaries under non-pathological conditions.

## Data Availability

The complete anonymized dataset supporting this experiment, comprising all 20 biochemical and hematological variables measured across six time points in seven exercised dromedary camels, has been uploaded with the manuscript as a Appendix A. The dataset includes individual animal time-series records corresponding to the analyses presented in the main text and Table 1. The full SAS code used for the repeated-measures ANOVA, Tukey-adjusted pairwise comparisons, and calculation of effect sizes (Cohen’s *d* with 95% confidence intervals) is provided as an accompanying Appendix A. All Appendix A are contained within a single compressed folder entitled “DATA and ANALYSIS.”.

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
