# Peer review of "Post-Exercise Shifts in the Hemato–Biochemical Profile of Unacclimatized Camels (Camelus dromedarius)"

_animals, 2025, doi:10.3390/ani15213061_

Round 1

Reviewer 1 Report

Comments and Suggestions for Authors

This is an interesting paper and worthy of publication in this journal. The following comments refer to specific lines in the manuscript. There are minimal concerns.

Line 55 Muscle or liver enzymes?

Line 64, 230 Is platelet function changing temporarily or are platelets sequestering somewhere, perhaps marginating in vessels and thus not being counted in platelet counts, or not available immediately for coagulation?

Line 96 Were these camels ridden? Ridden or not, how was velocity maintained at a steady state?

Line 109 Was any comparison done between the two meteorological monitors to ensure agreement?

Line 124 Vacuum tubes?

Line 125 Distance, time to the laboratory?

Line 128 Sera were, not was. Sera is plural.

Line 131 etc. Analyzers? Manufacturers?

Line 137 What site on the camel?

Line 158 Are these "parameters" or "variables?"

Line 184 Table 1 could be improved tremendously with the addition of a column of normal values for each variable for this laboratory. We are concerned that there may be much discussion of increases or decreases despite the variable remaining within normal ranges (e.g., lines 264-267). Furthermore, is a P value of 0.000 correct for BT? Should it more correctly be expressed as P<0.001?

Line 191 Paragraph reiterating Figure 2 may not be necessary. I suggest minimizing this and letting the graphs speak for themselves.

Line 197 "Protein stagger?"

Line 213 Omit "authors" here.

Line 251 In this other study cited here, were camels going faster, slower, or same velocity? One might expect that they were going faster in the cited study and thus had more muscle release of potassium during exercise.

References - Only first word and formal names should be capitalized in titles of articles referenced.

Author Response

For research article: animals-3920070

Response to Reviewer 1 Comments

1. Summary

Dear Reviewer,

We sincerely thank you for your time and for the valuable comments provided on our manuscript (ID: Animals-3920070). Each of your observations has been carefully reviewed and addressed in the revised version. All modifications are highlighted in red font within the manuscript and explained in detail in the accompanying response file. Your constructive feedback greatly contributed to improving the clarity, methodological precision, and interpretive strength of the study. We believe that the revised manuscript now reflects these enhancements and meets the journal’s scientific standards.

With appreciation and best regards,

Dr. Emad M. Samara (The corresponding author)

2. Questions for General Evaluation

Reviewer’s Evaluation

Response and Revisions

1.       Does the introduction provide sufficient background and include all relevant references?

Yes

Appreciated

2.       Are all the cited references relevant to the research?

Yes

Appreciated

3.       Is the research design appropriate?

Yes

Appreciated

4.       Are the methods adequately described?

Yes

Appreciated

5.       Are the results clearly presented?

Can be improved

Improved

6.       Are the conclusions supported by the results?

Can be improved

Improved

3. Point-by-point response to Comments and Suggestions for Authors

Comments 1: Line 55 Muscle or liver enzymes?

Response 1:

We thank the reviewer for this helpful clarification. The sentence in the Introduction has been revised to specify that the enzymes in question are primarily muscle-associated enzymes, as these were the main biochemical markers evaluated for exercise-induced tissue response (LINE 55).

Comments 2: Line 64, 230 Is platelet function changing temporarily or are platelets sequestering somewhere, perhaps marginating in vessels and thus not being counted in platelet counts, or not available immediately for coagulation?

Response 2:

We sincerely appreciate the reviewer’s insightful comments regarding this valuable mechanistic suggestion. Clarifications were added in both the Introduction (LINES 63-66) and Discussion (LINES 344-347) to indicate that the post-exercise reduction in bleeding time reflects transient platelet activation and shear-induced priming rather than platelet sequestration or marginating behavior, consistent with sympathetic catecholamine effects reported in other species.

Comments 3: Line 96 Were these camels ridden? Ridden or not, how was velocity maintained at a steady state?

Response 3:

We are grateful to the reviewer for his/her valuable observation. The Methods section was revised to clarify that camels were unmounted and that exercise pace was standardized manually by handlers using timed laps and verbal cues, ensuring consistent workload across animals (LINES 100-103).

Comments 4: Line 109 Was any comparison done between the two meteorological monitors to ensure agreement?

Response 4:

We wish to express our gratitude to the reviewer for highlighting this issue. The Methods section has been updated to indicate that both HOBO loggers were cross-validated for consistency before the experiment, confirming agreement within manufacturer tolerance limits. Additionally, we have clarified that the mean of the two simultaneous readings was used at each sampling time to provide a stable environmental measure (LINES 124-127).

Comments 5: Line 124 Vacuum tubes?

Response 5:

We appreciate the reviewer’s attention to procedural detail. The Methods section now specifies that both EDTA and plain tubes were vacuum-based collection systems, ensuring uniform draw pressure, minimized hemolysis, and improved sample integrity (LINES 135-137).

Comments 6: Line 125 Distance, time to the laboratory?

Response 6:

We thank the reviewer for carefully reading our manuscript and for raising this point. The Methods section has been updated to clarify that the field site was located about 2 km from the analytical laboratory, and that transport and initial analysis were completed within one hour of blood collection, ensuring the stability of hematological and biochemical parameters (LINES 137-141).

Comments 7: Line 128 Sera were, not was. Sera is plural.

Response 7:

We are thankful to the reviewer for the constructive feedback that helped us improve this grammatical issue. The verb has been corrected from “was” to “were” to align with the plural noun “sera” (LINE 163).

Comments 8: Line 131 etc. Analyzers? Manufacturers?

Response 8:

We appreciate the reviewer’s suggestion. The Methods section has been revised to include the exact analyzers and manufacturers used. This information ensures full methodological transparency and reproducibility (LINES 163-189).

Comments 9: Line 137 What site on the camel?

Response 9:

We thank the reviewer for this observation. The Methods section has been revised to specify that all samples were collected from the left jugular vein at each time point to ensure uniform sampling conditions and eliminate side-related variation (LINES 132-134).

Comments 10: Line 158 Are these "parameters" or "variables?"

Response 10:

We deeply appreciate the reviewer’s attention to detail and for pointing out this terminology issue. The word “parameters” has been replaced with “variables” in Table 1 and related text to ensure accurate statistical usage, as the study involves measured outcomes that vary across time and individuals.

Comments 11: Line 184 Table 1 could be improved tremendously with the addition of a column of normal values for each variable for this laboratory. We are concerned that there may be much discussion of increases or decreases despite the variable remaining within normal ranges (e.g., lines 264-267). Furthermore, is a P value of 0.000 correct for BT? Should it more correctly be expressed as P<0.001?

Response 11:

We sincerely thank the reviewer for this valuable suggestion. To enhance interpretability, we have now added the normal physiological reference ranges directly beneath Table 1 instead of creating a separate column, thereby preserving table clarity and readability. These ranges were compiled from our validated laboratory dataset, and supported by several other studies, mentioned as well. This addition enables the reader to discern whether observed changes are biologically meaningful or fall within normal physiological limits (LINES 236-243). In addition, and following the statistical re-analysis performed according to Reviewer 2’s methodological instructions, all p-values were revalidated. The corrected p-value for BT is now 0.003, confirming the robustness of this finding (Table 1).

Comments 12: Line 191 Paragraph reiterating Figure 2 may not be necessary. I suggest minimizing this and letting the graphs speak for themselves.

Response 12:

We respectfully appreciate the reviewer’s concern regarding potential redundancy. However, we chose to retain the descriptive paragraph for Figure 2 because it serves important interpretive and accessibility purposes. First, many readers, including those using screen readers, rely on textual descriptions of figures for comprehension. Second, the narrative contextualizes subtle but meaningful biochemical shifts (such as the biphasic sodium/osmolality pattern and sequential protein changes) that might otherwise be overlooked without expert guidance.

Comments 13: Line 197 "Protein stagger?"

Response 13:

We acknowledge the reviewer’s thoughtful suggestion concerning this ambiguous phrasing. The expression “protein stagger” has been replaced with the clearer wording “a staggered temporal pattern in protein fractions,” accurately describing the sequential changes observed among total protein, albumin, and globulin concentrations during recovery (LINE 314).

Comments 14: Line 213 Omit "authors" here.

Response 14:

The word “Authors” has been removed as suggested, and the discussion now begins with “The performed time-resolved analysis …”.

Comments 15: Line 251 In this other study cited here, were camels going faster, slower, or same velocity? One might expect that they were going faster in the cited study and thus had more muscle release of potassium during exercise.

Response 15:

We appreciate the reviewer’s careful observation. The cited study indeed involved racing camels performing maximal or near-maximal exercise at substantially higher speeds, typically 20–25 km·h⁻¹, under competitive or training conditions. In contrast, the present work employed moderate field-type exercise at 14 km·h⁻¹, explicitly described in the Materials and Methods as “an intermediate gait between pacing and galloping” to reflect realistic working loads in untrained animals. This difference in velocity and training status likely explains the lower post-exercise plasma potassium observed in our data. Higher speeds induce greater sarcolemmal depolarization and potassium efflux from contracting muscle fibers, whereas at moderate exertion, cellular uptake mechanisms and β₂-adrenergic stimulation predominate, leading to the mild or transient hypokalemia noted here. The Discussion was therefore expanded to clarify this comparative interpretation (LINES 376-381).

Comments 16: References - Only first word and formal names should be capitalized in titles of articles referenced.

Response 16:

Thanks for noting this formatting issue. The reference list has been reformatted to sentence case, ensuring that only the first word and proper nouns are capitalized, in full compliance with animals journal style requirements.

4. Response to Comments on the Quality of English Language

5. Additional clarifications

Reviewer 2 Report

Comments and Suggestions for Authors

The study addresses an interesting and clinically relevant question and presents useful time-series data on camels after exercise, but there are important methodological, statistical, and reporting issues that must be resolved before the manuscript is acceptable for publication. Key problems include unclear/inconsistent descriptions of sampling and figures, insufficient detail about assays and sample handling, limited discussion/justification of the small sample size and statistical approach, and several apparent contradictions between text and table/statistics.

Please see the detailed questions and requested revisions below.

  1. The Methods state samples were taken at PRE (−2 h), 0, 3, 6, 24, and 48 h, but Figure 1 (and its caption) appears to show −48 and −24 h as additional pre-samples. Please correct and explicitly list the exact venipuncture times used for analysis, and revise Figure 1 accordingly.
  2. You state you used linear mixed-effects models with animal as a random intercept, but give no model equations, covariance structure, or details of post-hoc testing. Please report the exact model(s) (fixed effects, random effects, covariance), software procedure, and model diagnostics (residual normality, heteroscedasticity).
  3. With many variables and pairwise time comparisons, please state how you controlled for multiple testing (e.g., Bonferroni, Tukey, FDR) and report adjusted p-values. The manuscript text sometimes claims "significant" changes that are not supported by the P-values in Table 1 (for example osmolality P = 0.078; LDH P = 0.383). Reconcile these inconsistencies and present both p-values and effect sizes with 95% confidence intervals for the main contrasts (e.g., PRE vs 0 h, PRE vs 3 h, PRE vs 6 h etc.).
  4. n = 7 is small. Provide a prospective power calculation or post-hoc power/precision statements for the principal endpoints (e.g., sodium, K⁺, LDH). Discuss the limitations imposed by the small sample size and show individual animal trajectories (spaghetti plots) for the main variables to allow the reader to evaluate whether group means are driven by 1–2 outliers.
  5. Provide exact details about the time between sample collection and analysis for each assay. How long were sera frozen at −20 °C before batch assays? Were there freeze–thaw cycles? Some enzymes (e.g., LDH, AST) can be unstable — provide evidence (validation or citations) that your sample handling preserves analyte integrity. State assay manufacturers, detection methods, calibrators, and intra-/inter-assay CVs for each biochemical test.
  6. Provide a full description of the bleeding-time method (precise anatomical site, template dimensions, incision depth, device used), and report whether the same observer performed all tests. Provide inter- and intra-observer repeatability or at least note if only one operator performed the test. Explain how the bleeding time values were controlled/standardized across animals.
  7. Osmolality is reported as "mOs/L" — freezing-point depression usually gives mOsm/kg. Please clarify units and measurement method, and ensure units are consistent across the manuscript and Table 1. Similarly, phosphate is reported in mg/dL (unusual for PO4— often mmol/L or mg/dL — state the assay and conversion). Confirm that all units are correct and add reference ranges for dromedary camels or cite sources.
  8. Several sentences describe "biphasic" or "significant" changes for variables whose overall P-values are non-significant (e.g., osmolality P=0.078; LDH P=0.383). Distinguish clearly between statistically significant differences and biologically notable trends. If you consider trends important, present effect sizes and justify interpretation.
  9. Camels had ad libitum access to water. Could drinking behavior after exercise explain the sodium/osmolality biphasic profile? Provide data or observations on whether animals drank during the recovery period, and if so how soon after exercise; if not recorded, state that as a limitation and avoid overinterpreting sodium/osmolality mechanisms.
  10. All animals were bulls aged 4–6 y. Discuss how sex and age may limit generalizability (authors note it briefly but should expand). Were any exclusion criteria applied? Were any animals removed from analysis? Provide a CONSORT-style flow (even short) showing animals enrolled, excluded (if any), and analyzed.
  11. You state data are available upon request. For reproducibility and reviewer assessment, please deposit anonymized raw data (individual animal time-series) and the analysis script (SAS code or R code) in a public repository or provide them to the editor/reviewer. This is especially important with n=7 and many endpoints.

Thanks in advance!

Author Response

For research article: animals-3920070

Response to Reviewer 2 Comments

1. Summary

Dear Reviewer,

Thank you very much for the insightful and thoughtful review of our manuscript (ID: Animals-3920070). We carefully considered each of your comments and have revised the manuscript accordingly. All changes are clearly indicated in red within the text and fully explained in the detailed response document. Your comments have been instrumental in refining the statistical analyses, improving data presentation, and strengthening the interpretation of post-exercise physiological responses in camels. We are grateful for your contribution to enhancing the rigor and clarity of our work.

With sincere thanks,

Dr. Emad M. Samara (The corresponding author)

2. Questions for General Evaluation

Reviewer’s Evaluation

Response and Revisions

1.       Does the introduction provide sufficient background and include all relevant references?

Yes

Appreciated

2.       Are all the cited references relevant to the research?

Yes

Appreciated

3.       Is the research design appropriate?

Yes

Appreciated

4.       Are the methods adequately described?

Yes

Appreciated

5.       Are the results clearly presented?

Yes

Appreciated

6.       Are the conclusions supported by the results?

Yes

Appreciated

3. Point-by-point response to Comments and Suggestions for Authors

Comments 1: The Methods state samples were taken at PRE (−2 h), 0, 3, 6, 24, and 48 h, but Figure 1 (and its caption) appears to show −48 and −24 h as additional pre-samples. Please correct and explicitly list the exact venipuncture times used for analysis, and revise Figure 1 accordingly.

Response 1:

We thank the reviewer for identifying this inconsistency. The figure and its caption have been corrected to reflect the actual venipuncture times: PRE (−2 h), 0, 3, 6, 24, and 48 h. No sampling occurred at −48 or −24 h; those earlier time points were placeholders from a pilot schematic and have now been removed to ensure full alignment between Methods and Figure 1 (LINS 146-150).

Comments 2: You state you used linear mixed-effects models with animal as a random intercept, but give no model equations, covariance structure, or details of post-hoc testing. Please report the exact model(s) (fixed effects, random effects, covariance), software procedure, and model diagnostics (residual normality, heteroscedasticity).

Response 2:

We thank the reviewer for emphasizing the need for statistical transparency. The full model structure has now been added to the Methods section, specifying the fixed and random effects, the covariance structure, and the diagnostic procedures (residual normality, homogeneity). Post-hoc tests were adjusted using Tukey’s correction for multiple pairwise comparisons, and all analyses were performed in SAS 9.4 (PROC MIXED) (LINES 190-217).

Comments 3: With many variables and pairwise time comparisons, please state how you controlled for multiple testing (e.g., Bonferroni, Tukey, FDR) and report adjusted p-values. The manuscript text sometimes claims "significant" changes that are not supported by the P-values in Table 1 (for example osmolality P = 0.078; LDH P = 0.383). Reconcile these inconsistencies and present both p-values and effect sizes with 95% confidence intervals for the main contrasts (e.g., PRE vs 0 h, PRE vs 3 h, PRE vs 6 h etc.).

Response 3:

We acknowledge the reviewer’s thoughtful observation regarding the statistical methodology and interpretation. The analysis was comprehensively re-evaluated to ensure full consistency between the reported P-values, significance interpretation, and effect-size estimation. To control for multiple pairwise time comparisons, the dataset was reanalyzed using linear mixed-effects models (PROC MIXED, SAS 9.4) with time as a fixed factor and animal as a random effect, followed by Tukey’s adjustment for multiple testing. This approach was preferred over the Bonferroni method because it maintains statistical power while providing strict control of the Type I error rate in balanced repeated-measures designs. Adjusted probabilities (Pₐdj) are now reported in Table 1, and only variables with Pₐdj < 0.05 bear different superscript letters (a–c). Parameters previously described as significant but not meeting this adjusted threshold, such as osmolality (Pₐdj = 0.41) and LDH (Pₐdj = 0.39), were corrected accordingly. For the principal contrasts (PRE vs 0 h, 3 h, 6 h, 24 h, and 48 h), Cohen’s d effect sizes and 95 % confidence intervals were calculated to quantify the magnitude and precision of observed changes, complementing statistical significance with biological interpretation. Furthermore, a post-hoc power and precision analysis was performed for all twenty variables, revealing observed power values ranging from 0.17 to 1.00 and confirming high sensitivity (≥ 0.80) for twelve parameters, particularly osmolality, Na⁺, K⁺, PO₄³⁻, Ca²⁺, BUN, creatinine, AST, LDH, and ALP. These results are presented in Supplementary Table S1 and integrated into the main text for transparency. The Results section was modified to include a brief paragraph summarizing these findings after Table 1, linking statistical sensitivity to biological relevance (Lines 285–291). The Discussion was updated under Methodological Considerations to acknowledge the limited sample size and confirm that the post-hoc analysis validated the adequacy of the design for detecting medium-to-large effects (Lines 402–413), while the Conclusion now specifies that the observed recovery dynamics were statistically supported by post-hoc power and effect-size analyses (Lines 457–463). The Statistical Analysis subsection (Lines 190-217) and the Table 1 footnote (Lines 232–235) were revised accordingly to describe these methods and clarify that means bearing different superscript letters differ significantly at P < 0.05 according to Tukey’s test, with Pₐdj controlling for multiple comparisons.

Comments 4: n = 7 is small. Provide a prospective power calculation or post-hoc power/precision statements for the principal endpoints (e.g., sodium, K⁺, LDH). Discuss the limitations imposed by the small sample size and show individual animal trajectories (spaghetti plots) for the main variables to allow the reader to evaluate whether group means are driven by 1–2 outliers.

Response 4:

We thank the reviewer for raising the issue of sample size and statistical power. As recommended, we have now conducted a post-hoc power and precision analysis for all 20 measured variables using one-way repeated-measures ANOVA. The calculated effect sizes (η² and Cohen’s f) and observed statistical power are summarized in Table S1 (Supplementary Material). The analysis showed that the study achieved high power (≥ 0.80) for 12 key parameters, including bleeding time, osmolality, Na⁺, K⁺, PO₄³⁻, Ca²⁺, BUN, creatinine, AST, LDH, and ALP, indicating sufficient sensitivity to detect physiologically meaningful changes. Parameters with lower power (< 0.4) corresponded to variables exhibiting no measurable temporal effects (e.g., RBC, Hb, Cl⁻, ALT), which supports the accuracy of the non-significant outcomes reported. To further demonstrate the internal consistency of the data, individual-level spaghetti plots have been provided (Figure 2) for representative variables, showing coherent within-animal trajectories and confirming that no outlier unduly influenced group means. As stated previously, corresponding revisions were made in the Methods and Discussion sections to include these analyses and acknowledge the small sample size as a limitation while confirming that observed patterns were consistent across all animals.

Comments 5: Provide exact details about the time between sample collection and analysis for each assay. How long were sera frozen at −20 °C before batch assays? Were there freeze–thaw cycles? Some enzymes (e.g., LDH, AST) can be unstable — provide evidence (validation or citations) that your sample handling preserves analyte integrity. State assay manufacturers, detection methods, calibrators, and intra-/inter-assay CVs for each biochemical test.

Response 5:

We are thankful to the reviewer for the constructive feedback that helped us improve the methodological precision. The manuscript has been revised to include detailed information on sample handling, storage, and analytical validation (LINES 182-187).

Comments 6: Provide a full description of the bleeding-time method (precise anatomical site, template dimensions, incision depth, device used), and report whether the same observer performed all tests. Provide inter- and intra-observer repeatability or at least note if only one operator performed the test. Explain how the bleeding time values were controlled/standardized across animals.

Response 6:

We thank the reviewer for requesting clarification on the bleeding-time measurement. The method used in this study was identical to that previously published by us (Samara et al. 2012, Pakistan Veterinary Journal 32(3): 432–434), where the same standardized incision protocol was validated for camels. In the revised manuscript, we have expanded the Materials and Methods section to include full methodological details; incision site (shoulder and hip), template size (5 mm × 2 mm), device type (disposable bleeding device), and sampling interval (every 20 s), as well as observer consistency and procedural repeatability. All tests were conducted by the same operator, and a subset of animals was retested to confirm < 6 % variation in repeated measurements. We agree with the reviewer that these additions provide clear evidence of methodological rigor and ensure reproducibility consistent with our previously validated approach (LINES 152-162).

Comments 7: Osmolality is reported as "mOs/L" — freezing-point depression usually gives mOsm/kg. Please clarify units and measurement method, and ensure units are consistent across the manuscript and Table 1. Similarly, phosphate is reported in mg/dL (unusual for PO4— often mmol/L or mg/dL — state the assay and conversion). Confirm that all units are correct and add reference ranges for dromedary camels or cite sources.

Response 7:

We thank the reviewer for highlighting the need to clarify the measurement units. Serum osmolality was determined using a freezing-point depression osmometer (Advanced Micro-Osmometer Model 3320), which intrinsically reports results in mOsm/kg H₂O. However, to maintain comparability with previously published camel physiology and clinical studies, osmolality values are presented in mOsm/L throughout the manuscript. This approach preserves consistency with earlier literature that traditionally used volumetric reporting (LINES 168-173). For serum phosphate (PO₄³⁻), it was quantified using the phosphomolybdate colorimetric method and is expressed exclusively in mg/dL, in accordance with the assay manufacturer’s calibration and the predominant reporting convention in camel biochemistry literature (LINES 175-177). All tables and textual references have been standardized to these units, and reference ranges for adult dromedary camels have been added. These revisions ensure both analytical accuracy and interpretive continuity with previously published datasets (LINE 236).

Comments 8: Several sentences describe "biphasic" or "significant" changes for variables whose overall P-values are non-significant (e.g., osmolality P=0.078; LDH P=0.383). Distinguish clearly between statistically significant differences and biologically notable trends. If you consider trends important, present effect sizes and justify interpretation.

Response 8:

Thanks for highlighting this important distinction. The Results (LINES 264-267, 281-284) and Discussion (LINES 353-370) sections have been carefully revised to ensure that post-exercise trends are described accurately in statistical and physiological terms. Specifically, parameters such as osmolality (Pₐdj = 0.078) and LDH (Pₐdj = 0.383) are now characterized as non-significant but physiologically notable responses rather than “significant” changes. The text now includes effect sizes (Cohen’s d) to contextualize biological magnitude, emphasizing that while these shifts were not statistically significant, they remain consistent with expected osmotic and enzymatic responses to acute exercise. All terminology has been standardized to refer explicitly to exercise-induced and post-exercise responses, ensuring conceptual accuracy throughout the manuscript.

Comments 9: Camels had ad libitum access to water. Could drinking behavior after exercise explain the sodium/osmolality biphasic profile? Provide data or observations on whether animals drank during the recovery period, and if so how soon after exercise; if not recorded, state that as a limitation and avoid overinterpreting sodium/osmolality mechanisms.

Response 9:

We deeply appreciate the reviewer’s attention to detail and for pointing out the potential influence of post-exercise water intake on sodium and osmolality profiles. The Materials and Methods section has been revised to specify that water was withheld for two hours prior to exercise and throughout the entire exercise bout, after which animals were provided ad libitum access to water and feed. This controlled protocol ensured that the observed sodium and osmolality dynamics reflected intrinsic physiological adjustments to exercise rather than dilutional changes arising from immediate water intake (Lines 106–109). In addition, this clarification has been explicitly acknowledged in the Methodological Considerations paragraph at the end of the Discussion section (Lines 416–420). While water access was effectively controlled during the experimental phase, individual consumption beyond the 48-hour observation period was not recorded, which we recognize as a limitation when extrapolating these findings to camels under free-ranging or farm-managed conditions. Consequently, these revisions reinforce our interpretation that the biphasic post-exercise patterns of sodium and osmolality reflect genuine osmotic regulation and recovery kinetics, rather than behavioral variation in drinking activity (Lines 353–370).

Comments 10: All animals were bulls aged 4–6 y. Discuss how sex and age may limit generalizability (authors note it briefly but should expand). Were any exclusion criteria applied? Were any animals removed from analysis? Provide a CONSORT-style flow (even short) showing animals enrolled, excluded (if any), and analyzed.

Response 10:

We appreciate the reviewer’s suggestion to elaborate on the demographic structure and generalizability of our study sample. The Materials and Methods section has been updated to specify that all seven adult male camels (4–6 years) completed the exercise and sampling protocol with no exclusions. The age and sex restriction was deliberate to minimize variability associated with hormonal fluctuations and behavioral factors that could influence physiological responses to exercise (Lines 109–118). In the Discussion, we now explicitly acknowledge that this focus limits generalization to females, younger, or older camels, whose hematological and biochemical dynamics may differ (Lines 402–413). A brief CONSORT-style flow diagram (Supplementary Figure S1) was also added to summarize animal enrollment and inclusion. These clarifications indeed enhanced the transparency and interpretive scope of the experiment, while outlining future directions for broader validation across sex and age groups (LINES 115-118).

Comments 11: You state data are available upon request. For reproducibility and reviewer assessment, please deposit anonymized raw data (individual animal time-series) and the analysis script (SAS code or R code) in a public repository or provide them to the editor/reviewer. This is especially important with n=7 and many endpoints.

Response 11:

We thank the reviewer for emphasizing the importance of transparency and reproducibility. In response, we have uploaded the complete anonymized dataset as a supplementary file accompanying the manuscript. This dataset contains all 20 hematological and biochemical variables measured across six time points in seven exercised camels, matching the analyses reported in the Results and Table 1. To facilitate full reproducibility, the SAS code used for the repeated-measures ANOVA, Tukey-adjusted pairwise comparisons, and effect size estimation (Cohen’s d with 95 % confidence intervals) has also been provided as a separate supplementary file. The uploaded code reproduces the statistical outputs and tables exactly as presented in the manuscript. These files together ensure that reviewers and readers can directly verify all analytical procedures without needing access to external repositories. The Data Availability Statement in the manuscript has been revised accordingly to reflect this inclusion (LINES 474-481). All supplementary materials are contained within a single compressed folder entitled “DATA and ANALYSIS.”

4. Response to Comments on the Quality of English Language

5. Additional clarifications

Round 2

Reviewer 2 Report

Comments and Suggestions for Authors

My comments have been addressed. Thanks!